# Seroprevalence of severe fever with thrombocytopenia syndrome using specimens from the Korea National Health & Nutrition Examination Survey

**Choon-Mee Kim[1], Mi Ah Han[2], Na Ra Yun[3], Mi-Seon Bang[3], You Mi Lee[3], Beomgi Lee[3], Dong-Min Kim[3]\***

**1** Premedical Science, College of Medicine, Chosun University, Gwangju, South Korea, **2** Department of Preventive Medicine, College of Medicine, Chosun University, Gwangju, South Korea, **3** Department of Internal Medicine, College of Medicine, Chosun University, Gwangju, South Korea

\* drongkim@chosun.ac.kr

## Abstract

### Background

Severe fever with thrombocytopenia syndrome (SFTS) is an acute febrile disease caused by bites from ticks infected with the SFTS virus. In Korea, SFTS patients are observed nationwide, including Jeju Island, but there are currently no data regarding the national prevalence of SFTS, including that of residents of 16 cities and provinces. This study aimed to investigate the seroprevalence of SFTS in Korea.

### Methodology/principal findings

A total of 1500 participants were selected through random sampling from the 2014–2015 Korea National Health and Nutrition Examination Survey (KNHANES). An indirect immuno-fluorescence antibody assay (IFA) was performed to assess immunoglobulin (Ig) G and IgM antibody titers against SFTS virus.

### Results

Of the 1500 participants, 55 (3.7%) tested positive for IgG and 1 (0.1%) tested positive for IgM, with antibody titer of $\geq$ 1:32. Approximately 3.9% and 2.5% of participants in urban and rural areas, respectively, had a positive titer of $\geq$ 1:32. There was a significant correlation between SFTS incidence per 100,000 population and seroprevalence using an IgG titer $\geq$ 1:64 as the cut-off value.

### Conclusion

This is the first study to investigate national SFTS seroprevalence in all 16 cities and provinces representing Korea. Our study will also provide useful guidelines for the development of preventive measures against SFTS.

**Data Availability Statement:** All relevant data are within the manuscript and its Supporting Information files.

**Funding:** C-MK received grant supported by the Research Program funded by the Korea Disease Control and Prevention Agency (fund code 2017P2300300). This study was supported by research fund (team research project) from Chosun University 2022. The funder had no role in study design, data collection and analysis, decision to publish, or preparation of the manuscript.

**Competing interests:** The authors have declared that no competing interests exist.

## Author summary

Severe fever with thrombocytopenia syndrome (SFTS) is an acute febrile disease caused by a bite from a tick carrying the SFTS virus. This new infectious disease is observed mainly in Korea, China, and Japan. In Korea, SFTS incidence reported in Korea Disease Control and Prevention Agency is an average of 233 cases between 2016 and 2020, with 39 deaths and a fatality rate of 16.68%. SFTS patients are observed nationwide, including Jeju Island, but there are currently no data regarding the national prevalence of SFTS, including that of residents of 16 cities and provinces. In the present study, we aimed to investigate the seroprevalence of SFTS in Korea. A total of 1500 participants were selected through random sampling from the 2014–2015 Korea National Health and Nutrition Examination Survey. An indirect immunofluorescence antibody assay was performed to assess immunoglobulin (Ig) G and IgM antibody titers against SFTS virus. Of the 1500 participants, 55 (3.7%) tested positive for IgG and 1 (0.1%) tested positive for IgM, with antibody titer of $\geq$ 1:32. Approximately 3.9% and 2.5% of participants in urban and rural areas, respectively, had a positive titer of $\geq$ 1:32. There was a significant correlation between SFTS incidence per 100,000 population and seroprevalence using an IgG titer $\geq$ 1:64 as the cut-off value. This is the first study to investigate national SFTS seroprevalence across all 16 cities and provinces representing Korea; furthermore, our study will also provide useful guidelines for the development of preventive measures against SFTS.

## Introduction

Severe fever with thrombocytopenia syndrome (SFTS) is an acute febrile disease caused by a bite from a tick carrying the SFTS virus. This syndrome is characterized by symptoms including decreased leukocyte and platelet levels, elevated liver enzyme levels, and proteinuria [1]. SFTS was first reported in China in 2011 [2]. In Korea, among the five groups of legal infectious diseases, SFTS is a Group 3 infectious disease. Furthermore, since the first case was identified in 2013, its incidence has increased annually up to 2020 [3]. Group 3 infectious diseases include malaria, tuberculosis, scrub typhus, and SFTS. These diseases may prevail intermittently; therefore, continuous monitoring of outbreaks and the establishment of preventive measures are required (https://www.kdca.go.kr/npt/biz/npp/portal/nppLwcrIcdMain.do). SFTS is a new infectious disease that occurs mainly in Korea, China, and Japan. The Korea Disease Control and Prevention Agency has reported an average of 233 cases in the past 5 years (between 2016 and 2020), with 39 deaths and a fatality rate of 16.68% (http://www.kdca.go.kr/npt/biz/npp/ist/simple/simplePdStatsMain.do).

In China, systematic reviews and meta-analyses to evaluate STFS seroprevalence were investigated up to November 2016, and an overall pooled antibody positivity rate of 4.3% was reported [4]. In SFTSV antibody testing on 2,510 residents in Jiangsu Province, China, 0.44% of the residents tested positive [5]. However, the seropositivity rate for farmers in rural areas of the same province was 1.3%, which was three times higher than that of the general population [6, 7]. In addition, the seroprevalence of SFTSV infections was tested in 1,375 healthy persons from Penglai County, eastern China, and the serum of 3.85% of the general population tested positive for SFTSV antibodies [8]. In contrast, 7.2% of healthy residents in Zhejiang Province, China were positive for SFTSV IgG antibodies [9]. According to a report in 2018 that examined SFTS seroprevalence, 0.3% (2/646) of healthy individuals residing in Kagoshima, Japan, were positive for anti-SFTS antibodies. Moreover, 1,000 serum samples collected from general

blood donors by the Japanese Red Cross Kyushu Block Blood Center were negative for anti-SFTSV antibodies, suggesting a low seroprevalence of SFTS antibodies in individuals living in endemic areas in Japan [10].

Our group previously assessed antibody titers according to regional and residential characteristics, tick contact history, and occupational and outdoor activity-related characteristics by investigating antibody positivity rates in residents of high-risk areas. Fifty (4.1%) of the 1,228 residents of rural areas exhibited an SFTS antibody titer of > 1:32, of which 23 (5.5%) in Boseong, 16 (4.0%) in Hapcheon, and 11 (2.7%) in Dangjin were seropositive for the SFTSV [11]. In addition, in a study conducted on 1,069 serum samples from patients who visited a tertiary hospital in southeastern Korea, 2.1% of the patients were positive for anti-SFTSV antibodies, of which 7.7% of patients from rural areas and 1.9% of patients from urban areas were shown to be seropositive [6].

To date, most studies investigating SFTS antibody positivity rates have been conducted in regions with a high incidence of disease. In addition, because there are large variations across geographical regions, the findings were insufficient to be representative of national SFTS antibody positivity rates. Therefore, the purpose of this study was to investigate nationwide SFTS antibody positivity rate using serum samples collected in the Korea National Health and Nutrition Examination Survey (KNHANES) and compare it with SFTS incidence rate per 100,000 population.

## Methods

### Ethics statement

This study was approved by the Institutional Review Board (IRB) of Chosun University Hospital (IRB No. 2017-12-014). Written informed consent was obtained from all participants for the use of blood samples in Korea National Health and Nutrition Examination Survey. Written informed consent was obtained from the parent/guardian/head of household in case of participants less than 18 years old.

### Specimen selection

To assess SFTS seroprevalence in samples collected in the 2014–2015 KNHANES by the National Biobank of the Korea Disease Control and Prevention Agency (KDCA), 1500 participants aged 10–80 years were randomly selected by random sampling from all 16 cities and provinces in Korea based on the number of individuals in each province, as well as their age and sex. We received 1,500 stored serum samples collected for KNHANES from the National Biobank of KDCA in two batches (800 samples for the first (March 30, 2018) and 700 samples for the second (April 06, 2018)), and stored them at -20˚C until use.

### Diagnostic method

An indirect immunofluorescence antibody assay (IFA) was performed to assess immunoglobulin (Ig) G and IgM antibody titers against the SFTS virus. To prepare SFTS antigen slides, STFS virus-infected Vero E6 cells were plated on Teflon-coated well slides, incubated overnight, and fixed using 80% acetone. Serum samples collected in KNHANES were serially diluted two-fold and were used as primary antibodies to react with viral antigens in a humid chamber at 37˚C for 30 min. After washing with PBS and distilled water, 1:400 dilutions of fluorescein isothiocyanate-conjugated anti-human immunoglobulin (Ig) G and IgM (MP Biomedicals, Illkirch, France) were added to each slide as secondary antibodies and incubated at 37˚C for 30 min in a humid chamber. Antifade mounting medium (Vector Laboratories,

Burlingame, CA, USA) was then dispensed, and the slides were observed under a fluorescence microscope (Olympus IX73, magnification 400×). Antibody titers were determined using final serum dilutions that exhibited specific fluorescence signatures. IFA IgG and IgM antibody titers ≥ 1:32 were defined as positive cut-off values based on results from 15 health check-up participants [11].

## Statistical analysis

A list of variables for individual characteristics of participants and database (DB) for each variable was obtained from the KNHANES, and the results were analyzed and compared with the SFTS antibody titer results. Patient characteristics are expressed as frequency and percentage, as well as median and quartiles. The Chi-square test or Fisher's exact test was used to assess the difference between the investigated risk factors and SFTS. Spearman correlation analysis was used to assess the correlation between the number of SFTS cases per 100,000 population and SFTS IFA antibody positivity rate in 16 regions from 2014 to 2015. Differences were considered statistically significant at $p < 0.05$. Statistical analysis was performed using the Statistical Package for the Social Sciences (SPSS), v24.0 software (IBM Corp., Armonk, USA).

## Results

### Sample distribution

Of the 1,500 participants, the greatest numbers of participants were from Gyeonggi and Seoul, with 23.2% (348) and 19.7% (295), respectively (Fig 1A). Of the participants, 50.1% (751) were men and 49.9% (749) were women. Approximately 3.0% (n = 45) of the participants were between 10 and 19 years of age, with 12.0% (n = 180), 15.0% (n = 225), 20.0% (n = 300), 20.0% (n = 300), and 30.0% (n = 450) between 20 and 29, 30–39, 40–49, 50–59, and > 60 years of age, respectively (Fig 1B). When the participants were grouped according to urban (*dong*) and rural (*eup/myun*) areas, 81.2% (1218) were from urban areas, and 18.8% (282) were from rural areas. The number of participants according to age distribution and regions, the population of each region in 2014, and the ratio of participants to the population (%) are shown in S1 and S2 Tables.

### IFA Results

Fifty-five of 1,500 participants (3.7%) were seropositive for the SFTS virus with an IgG antibody titer of ≥ 1:32 (S1 Fig). The SFTS IgM antibody titer was ≥ 1:32 in 1 of 1500 participants (0.1% positivity rate). The IFA IgG antibody titer was ≥ 1:64, ≥ 1:128, and ≥ 1:256 in 1.8% (27/1500), 0.6% (9/1500), and 0.1% (1/1500) of participants, respectively. The final antibody titer of one participant who tested positive for SFTS IFA IgM was 1:64 (Table 1 and S2 Fig).

We investigated regional cases of SFTS patients from 2014 to 2015, reported in the Infectious Disease Portal of the KDCA (http://www.kdca.go.kr/npt/biz/npp/ist/simple/simple.PdStatsMain.do). The annual incidence of SFTS was 55 in 2014 and 79 in 2015. The greatest number of patients was observed in Gyeongbuk (n = 28), followed by Gangwon (n = 19), Jeju (n = 16), Gyeonggi (n = 15), and Gyeongnam (n = 15). However, the regional incidence of SFTS per 100,000 was the highest in Jeju, with 2.66 cases per 100,000 (Fig 2).

The SFTS regional seroprevalence based on a cutoff titer of ≥1:32 was Seoul 3.1%, Busan 4.8%, Daegu 7.5%, Incheon 7.4%, Gwangju 1.9%, Daejeon 3.7%, Ulsan 3.0%, Gyeonggi-do 2.6%, Gangwon-do 8.0%, Chungcheongbuk-do 0%, Chungcheongnam-do 1.6%, Jeollabuk-do 3.6%, Jeollanam-do 0%, Gyeongsangbuk-do 5.5%, Gyeongsangnam-do 2.5%, and Jeju 6.3%,

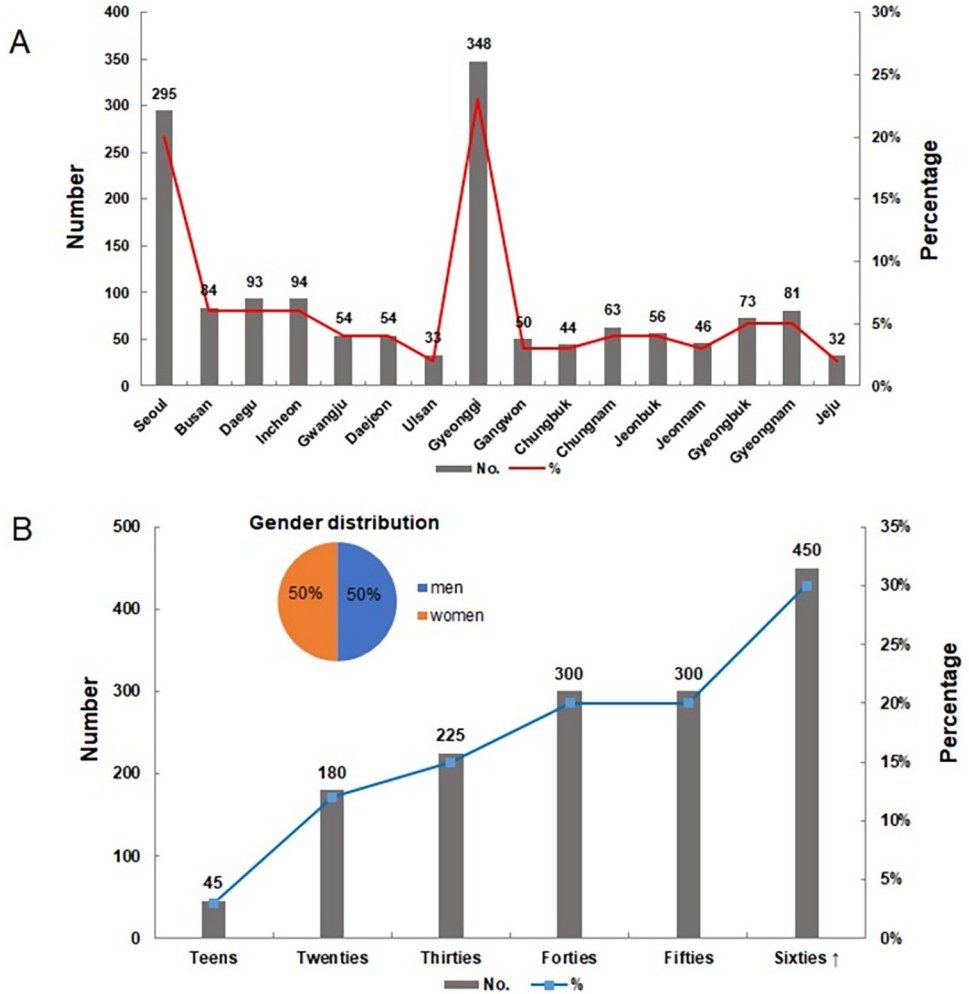

**Fig 1.** Distribution of the 1500 participants selected from the Korea National Health and Nutrition Examination Survey, 2014 to 2015 according to region (A), and age and sex (B). Gyeonggi, Gyeonggi-do; Gangwon, Gangwon-do; Chungbuk, Chungcheongbuk-do; Chungnam, Chungcheongnam-do; Jeonbuk, Jeollabuk-do; Jeonnam, Jeollanam-do; Gyeongbuk, Gyeongsangbuk-do; Gyeongnam, Gyeongsangnam-do; Jeju, Jeju-do.

respectively (Fig 3A). When we divided the patients into urban and rural areas, the SFTS seroprevalence was 3.9% (48/1,218) for urban areas and 2.5% (7/282) for rural areas.

The incidence of SFTS per 100,000 and IgG antibody positivity rates with titers ≥ 1:32 and ≥ 1:64 per region were compared. In Jeju, the incidence of SFTS was 2.66 cases per

**Table 1. SFTS seropositivity determined by IFA in samples collected from the Korea National Health and Nutrition Examination Survey, 2014 to 2015.**

| Antibody titer | IFA IgG | | IFA IgM | |
|---|---|---|---|---|
| | Number (%) | Cumulative percent (%) | Number (%) | Cumulative percent (%) |
| 1:256 | 1 (0.1) | 0.1 | 0 (0) | 0.0 |
| 1:128 | 8 (0.5) | 0.6 | 0 (0) | 0.0 |
| 1:64 | 18 (1.2) | 1.8 | 1 (0.1) | 0.1 |
| 1:32 | 28 (1.9) | 3.7 | 0 (0.1) | 0.1 |
| < 1:32 | 1,445 (96.3) | - | 1,499 (99.9) | - |
| Total Number | 1,500 (100.0) | | 1,500 (100.0) | |

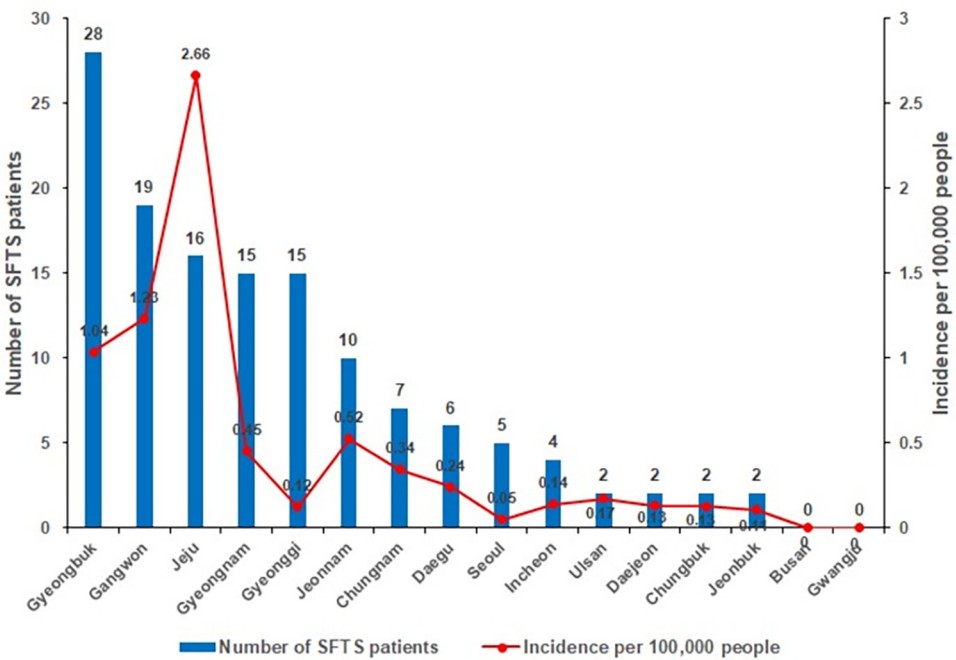

**Fig 2. Number of severe fever with thrombocytopenia syndrome (SFTS) patients reported in Infectious Disease Portal of Korea Disease Control and Prevention Agency according to region and incidence per 100,000 between 2014 and 2015.**

100,000, and SFTS IgG antibody positivity rate with titer $\geq$ 1:32 was 6.3%; in Gangwon-do, SFTS incidence was 1.23 cases per 100,000, and SFTS IgG antibody positivity with titer $\geq$ 1:32 was 8.0%. The SFTS IgG antibody positivity rate with titer $\geq$ 1:64 was highest in Jeju and Gangwon-do (6.3% and 6.0%, respectively) (Fig 3B).

The 16 cities and provinces investigated in KNHANES were divided into seven metropolitan centers (Seoul, Busan, Daegu, Incheon, Gwangju, Daejeon, and Ulsan) and nine provinces (Gyeonggi-do, Gangwon-do, Chungcheongbuk-do, Chungcheongnam-do, Jeollabuk-do, Jeollanam-do, Gyeongsangbuk-do, Gyeongsangnam-do, and Jeju-do). Overall SFTS seropositive rates with titers $\geq$ 1:32 were observed in 4.4% (31/707) and 3.0% (24/793) of participants from metropolitan centers and provinces, respectively.

Spearman correlation analysis was performed to assess the association between regional antibody positivity rates and the number of SFTS patients according to region, as well as SFTS incidence per 100,000 between 2014 and 2015. There was no significant relationship between SFTS seroprevalence with titer $\geq$ 1:32 and the number of SFTS patients or incidence per 100,000. However, SFTS incidence per 100,000 was significantly correlated with SFTS IgG seroprevalence with a titer $\geq$ 1:64 (r = 0.723, p = 0.002) (Fig 4).

## Factors related to SFTS antibody positivity

Antibody positivity rates were compared based on the general characteristics of the participants. Among the participants in the 2014–2015 KNHANES, 2.9% (22/751) and 4.4% (33/749) of men and women were positive for SFTS antibodies, respectively; however, there was no significant difference between the two groups. In addition, the SFTS antibody positivity rate was not significantly different according to age, marital status, income level, and education level (Table 2).

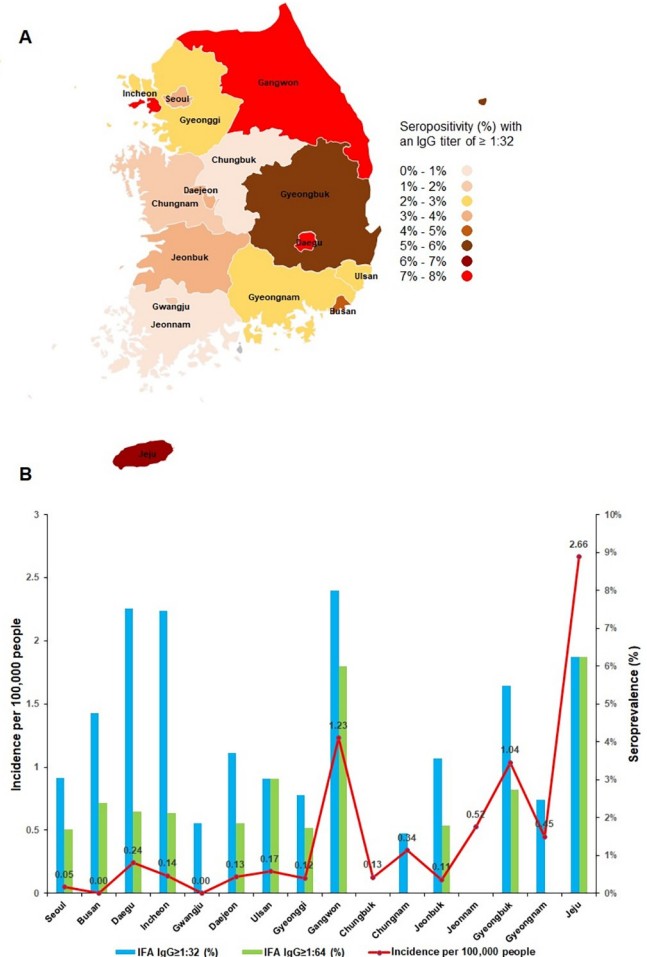

**Fig 3. Regional incidence of severe fever with thrombocytopenia syndrome (SFTS) per 100,000 individuals for each of the 16 cities and provinces, and immunoglobulin (Ig) G antibody positivity rate.** A. Choropleth map showing national SFTS seroprevalence (%) with IgG titer of ≥ 1:32. Choropleth map was amended using base map obtained from "Editable South Korea Map Template For Presentation" (https://www.slideegg.com/powerpoint/south-korea-map-powerpoint-templates). B. Plot showing regional SFTS incidence per 100,000 individuals and IgG positivity rate (titer ≥ 1:32, ≥ 1:64).

SFTS antibody positivity rate was further analyzed according to underlying diseases, influenza vaccination, economic activity, and occupational characteristics (of those employed) of the participants. The SFTS seropositivity rate with titers ≥ 1:128 was significantly higher in participants with underlying diseases (1.6%) than in those without underlying diseases (0.1%) (p = 0.001). In particular, the antibody positivity rate was significantly correlated with the presence of cardiovascular disease among underlying diseases (e.g., hypertension, dyslipidemia, stroke, myocardial infarction, angina, and renal failure) (p = 0.018). In addition, there was a significant correlation between SFTS seroprevalence with titers ≥ 1:128 and influenza vaccination (p = 0.0003) (Table 3).

## Discussion

SFTS is an acute febrile disease caused by the SFTS virus. In Korea, it is observed nationwide, including Jeju Island, and is a fatal disease associated with a high mortality rate of 10.9% [12, 13].

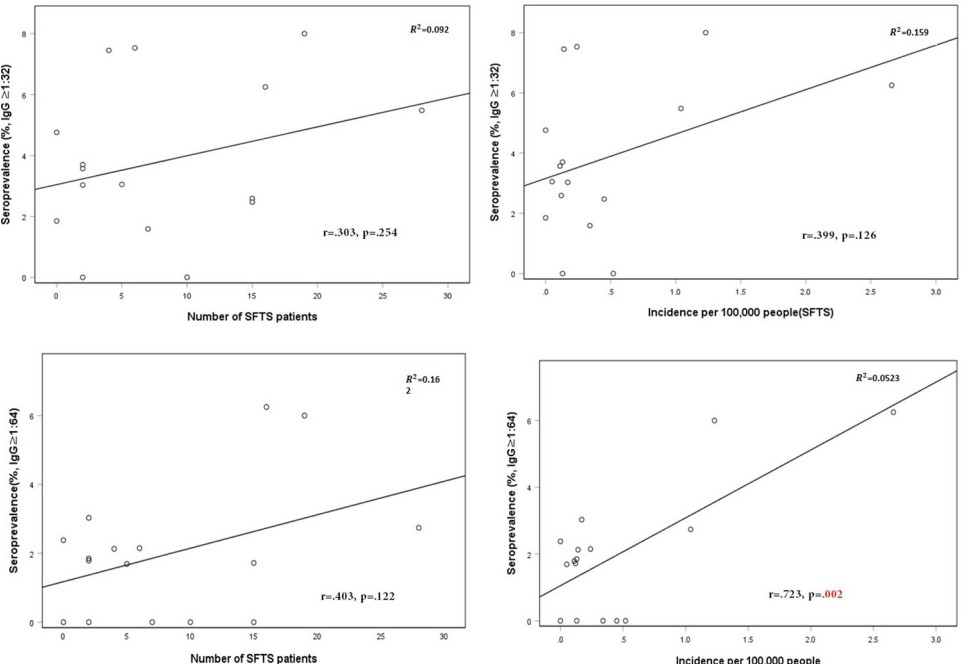

**Fig 4.** Correlation of the number of severe fever with thrombocytopenia syndrome (SFTS) patients (left) and SFTS incidence per 100,000 (right) with IgG antibody positivity rates (IgG titer, $\geq$ 1:32, A; IgG titer $\geq$ 1:64, B) between 2014 and 2015.

SFTS seroprevalence is an indicator of infection that can be used to determine the spread of SFTS in the community and an indicator of immunity against SFTS infection. Because guidelines and levels of response to infectious diseases may change depending on the antibody seropositive rate, information regarding the seroprevalence of SFTS antibodies is important in responding to outbreaks of infectious diseases.

This study investigated the SFTS seroprevalence using 1500 samples collected in KNHANES across the national population using an antibody titer of 1:32 as the cut-off, IFA IgG and IgM positivity rates of 3.7% and 0.1%, respectively, were observed.

In our previous study, the SFTS antibody positivity rate in three high-risk areas in Korea was 4.1% [11]. In the current study, the antibody positivity rate based on 1500 samples from KNHANES in 16 cities and provinces nationwide in Korea was 3.7%. Although direct comparison of the antibody positivity rate between these two studies is difficult, nationwide seroprevalence using KNHANES samples was also high. In particular, seroprevalence was not significantly different between participants from urban (3.9%) and rural (2.58%) areas, suggesting that urban residents may not be safe from SFTS infection. When the antibody positivity rate was compared with the number of SFTS patients and incidence per 100,000, similar patterns were observed despite regional differences. In particular, using an antibody titer cut-off value $\geq$ 1:64, SFTS incidence per 100,000 was significantly correlated with SFTS IgG seroprevalence.

According to a study on SFTS seroprevalence in Xinyang—the most severe SFTS-endemic region of China—the seropositive rate was 6.59%. In addition, the SFTS seroprevalence was higher in individuals who lived in rural areas and in people that were sampled in post-endemic seasons, suggesting that the seroprevalence of SFTSV-specific IgG antibodies was significantly correlated with the reported incidence rate and temporal and geographic level [14].

**Table 2.** SFTS seropositivity and epidemiological characteristics of the 1500 participants selected from the Korea National Health and Nutrition Examination Survey, 2014 to 2015.

| Classification | Number | IFA (titer≥1:32) | | IFA (titer≥1:64) | | IFA (titer≥1:128) | | IFA (titer≥1:256) | |
|---|---|---|---|---|---|---|---|---|---|
| | | N (%) | P-value | N (%) | P-value | N (%) | P-value | N (%) | P-value |
| Sex | | | 0.128 | | 0.555 | | 0.753 | | 1.000 |
| Men | 751 | 22 (2.9) | | 12 (1.6) | | 4 (0.5) | | 1 (0.1) | |
| Women | 749 | 33 (4.4) | | 15 (2.0) | | 5 (0.7) | | 0 (0.0) | |
| Age | | | 0.122 | | 0.204 | | 0.383 | | 0.094 |
| 10s | 45 | 3 (6.7) | | 0 (0.0) | | 0 (0.0) | | 0 (0.0) | |
| 20s | 180 | 9 (5.0) | | 5 (2.8) | | 1 (0.6) | | 0 (0.0) | |
| 30s | 225 | 13 (5.8) | | 7 (3.1) | | 1 (0.4) | | 0 (0.0) | |
| 40s | 300 | 11 (3.7) | | 5 (1.7) | | 2 (0.7) | | 0 (0.0) | |
| 50s | 300 | 5 (1.7) | | 4 (1.3) | | 1 (0.3) | | 0 (0.0) | |
| >60s | 450 | 14 (3.1) | | 6 (1.3) | | 4 (0.9) | | 1 (0.2) | |
| Marital status | | | 0.535 | | 0.952 | | 0.649 | | 1.000 |
| Single | 279 | 13 (4.7) | | 6 (2.2) | | 1 (0.4) | | 0 (0.0) | |
| Married | 1,056 | 35 (3.3) | | 18 (1.7) | | 8 (0.8) | | 1 (0.1) | |
| Others | 165 | 7 (4.2) | | 3 (1.8) | | 0 (0.0) | | 0 (0.0) | |
| Income | | | 0.508 | | 0.429 | | 0.541 | | 0.401 |
| Lowest | 228 | 9 (3.9) | | 2 (0.9) | | 1 (0.4) | | 0 (0.0) | |
| Lower middle | 371 | 10 (2.7) | | 7 (1.9) | | 4 (1.1) | | 1 (0.3) | |
| Higher middle | 416 | 14 (3.4) | | 6 (1.4) | | 1 (0.2) | | 0 (0.0) | |
| Highest | 478 | 22 (4.6) | | 12 (2.5) | | 3 (0.6) | | 0 (0.0) | |
| Education | | | 0.937 | | 0.813 | | 0.617 | | 0.635 |
| Elementary or less | 241 | 10 (4.1) | | 3 (1.2) | | 1 (0.4) | | 0 (0.0) | |
| Middle school | 162 | 6 (3.7) | | 2 (1.2) | | 2 (1.2) | | 0 (0.0) | |
| High school | 524 | 18 (3.4) | | 11 (2.1) | | 4 (0.8) | | 0 (0.0) | |
| University or higher | 509 | 21 (4.1) | | 11 (2.2) | | 2 (0.4) | | 1 (0.2) | |
| Urban/rural classification | | | 0.240 | | 0.126 | | 0.68 | | 1.000 |
| Urban area | 1218 | 48 (3.9) | | 25 (2.1) | | 7 (0.6) | | 1 (0.1) | |
| Rural areas | 282 | 7 (2.5) | | 2 (0.7) | | 2 (0.7) | | 0 (0.0) | |
| House type | | | 0.100 | | 0.079 | | 0.101 | | 1.000 |

(*Continued*)

**Table 2.** (Continued)

| Classification | Number | IFA (titer≥1:32) | | IFA (titer≥1:64) | | IFA (titer≥1:128) | | IFA (titer≥1:256) | |
|---|---|---|---|---|---|---|---|---|---|
| | | N (%) | P-value | N (%) | P-value | N (%) | P-value | N (%) | P-value |
| House | 548 | 20 (3.6) | | 8 (1.5) | | 4 (0.7) | | 0 (0.0) | |
| Apartment | 926 | 32 (3.5) | | 17 (1.8) | | 4 (0.4) | | 1 (0.1) | |
| Others | 26 | 3 (11.5) | | 2 (7.7) | | 1 (3.8) | | 0 (0.0) | |
| Household type | | | 0.609 | | 0.453 | | 0.376 | | 0.322 |
| Single person | 134 | 3 (2.2) | | 1 (0.7) | | 0 (0.0) | | 0 (0.0) | |
| One generation | 349 | 12 (3.4) | | 5 (1.4) | | 4 (1.1) | | 1 (0.3) | |
| >2 generations | 1,017 | 40 (3.9) | | 21 (2.1) | | 5 (0.5) | | 0 (0.0) | |
| Employment status | | | 0.985 | | 0.793 | | 0.29 | | 1.000 |
| Unemployed | 496 | 19 (3.8) | | 10 (2.0) | | 5 (1.0) | | 0 (0.0) | |
| Employed | 935 | 36 (3.9) | | 17 (1.8) | | 4 (0.4) | | 1 (0.1) | |
| Occupation | | | 0.641 | | 0.319 | | 0.725 | | 1.000 |
| Agricultural, forestry, and fisheries | 80 | 2 (2.5) | | 0 (0.0) | | 0 (0.0) | | 0 (0.0) | |
| Management, office, and service | 556 | 24 (4.3) | | 14 (2.5) | | 3 (0.5) | | 1 (0.2) | |
| Technical, machine, and production | 163 | 4 (2.5) | | 1 (0.6) | | 0 (0.0) | | 0 (0.0) | |
| Others | 133 | 6 (4.5) | | 2 (1.5) | | 1 (0.8) | | 1 (0.0) | |
| Total | | 55 (3.7) | | 27 (1.8) | | 9 (0.6) | | 1 (0.1) | |

In a cohort study of a healthy population in a highly endemic region of China, the overall seropositive rates were 11.9% (IgG antibody) and 6.8% (neutralizing antibody), respectively. Moreover, it was reported that anti-SFTSV IgG seropositivity was significantly related to older age (≥70 years), recent contact with cats, and working in tea gardens [15].

In 2015, a cross-sectional study was performed on 1069 serum samples obtained from university hospitals in the southeastern region of Korea [6]. Double-antigen sandwich ELISA testing using nucleoproteins revealed that 22 (2.1%) patients were seropositive for the SFTS virus. SFTS seropositivity increased with age (p = 0.034), and the SFTS antibody positivity rate in rural areas was 7.7% (2/26), which was higher than that in urban areas (1.9% [20/1043]; p = 0.040).

The antibody positivity rate was not significantly different according to the underlying disease(s) [6]. However, in our study, we did not observe such significant differences in antibody positivity rates between the urban and rural regions. We also assessed factors related to SFTS antibody positivity according to the epidemiological characteristics of the 2014–2015 KNHANES participants. The seroprevalence of SFTS antibody titers ≥ 1:128 was significantly higher according to the presence of underlying diseases, especially cardiovascular diseases (e.g., hypertension, dyslipidemia, stroke, myocardial infarction, angina, and renal failure). In

**Table 3. SFTS seropositivity according to underlying diseases, influenza vaccination, economic activity, and occupational characteristics of the 1500 participants selected from the Korea National Health and Nutrition Examination Survey, 2014 to 2015.**

| Classification | Number | IFA (titer≥1:32) | | IFA (titer≥1:64) | | IFA (titer≥1:128) | | IFA (titer≥1:256) | |
|---|---|---|---|---|---|---|---|---|---|
| | | N (%) | *P*-value | N (%) | *P*-value | N (%) | *P*-value | N (%) | *P*-value |
| Underlying diseases | | | 0.326 | | 0.551 | | **0.001** | | 0.353 |
| No | 942 | 39 (4.1) | | 16 (1.7) | | 1 (0.1) | | 0 (0.0) | |
| Yes | 514 | 16 (3.1) | | 11 (2.1) | | 8 (1.6) | | 1 (0.2) | |
| Cardiovascular diseases | | | 0.447 | | 0.864 | | **0.018** | | 0.282 |
| No | 1046 | 42 (4.0) | | 19 (1.8) | | 3 (0.3) | | 0 (0.0) | |
| Yes | 410 | 13 (3.2) | | 8 (2.0) | | 6 (1.5) | | 1 (0.2) | |
| Infectious diseases | | | 0.218 | | 0.067 | | 0.135 | | 1.000 |
| No | 1418 | 53 (3.7) | | 25 (1.8) | | 8 (0.6) | | 1 (0.1) | |
| Yes | 23 | 2 (8.7) | | 2 (8.7) | | 1 (4.3) | | 0 (0.0) | |
| Hematooncologic diseases | | | 0.721 | | 0.33 | | 0.301 | | 1.000 |
| No | 1385 | 54 (3.9) | | 26 (1.9) | | 8 (0.6) | | 1 (0.1) | |
| Yes | 56 | 1 (1.8) | | 1 (1.8) | | 1 (1.8) | | 0 (0.0) | |
| Rheumatoid arthritis | | | 0.561 | | 1 | | 0.124 | | 1.000 |
| No | 1420 | 54 (3.8) | | 26 (1.8) | | 8 (0.6) | | 1 (0.1) | |
| Yes | 21 | 1 (4.8) | | 1 (4.8) | | 1 (4.8) | | 0 (0.0) | |
| Diabetes | | | 0.618 | | 1 | | 0.157 | | 0.080 |
| No | 1340 | 52 (3.9) | | 25 (1.9) | | 7 (0.5) | | 0 (0.0) | |
| Yes | 116 | 3 (2.6) | | 2 (1.7) | | 2 (1.7) | | 1 (0.9) | |
| Others | | | 1.000 | | 0.622 | | 1.000 | | 1.000 |
| No | 1335 | 53 (4.0) | | 27 (2.0) | | 9 (0.7) | | 1 (0.1) | |
| Yes | 53 | 2 (3.8) | | 0 (0.0) | | 0 (0.0) | | 0 (0.0) | |
| Influenza vaccination | | | 0.580 | | 0.286 | | **0.003** | | 0.383 |
| No | 889 | 32 (3.6) | | 14 (1.6) | | 1 (0.1) | | 0 (0.0) | |
| Yes | 551 | 23 (4.2) | | 13 (2.4) | | 8 (1.5) | | 1 (0.2) | |
| Employment status | | | 0.985 | | 0.793 | | 0.29 | | 1.000 |
| Unemployed | 496 | 19 (3.8) | | 10 (2.0) | | 5 (1.0) | | 0 (0.0) | |
| Employed | 935 | 36 (3.9) | | 17 (1.8) | | 4 (0.4) | | 1 (0.1) | |
| Occupation | | | 0.770 | | 0.219 | | 1.000 | | 1.000 |
| Office/service | 556 | 24 (4.3) | | 14 (2.5) | | 3 (0.5) | | 1 (0.2) | |

(*Continued*)

**Table 3.** (Continued)

| Classification | Number | IFA (titer≥1:32) | | IFA (titer≥1:64) | | IFA (titer≥1:128) | | IFA (titer≥1:256) | |
|---|---|---|---|---|---|---|---|---|---|
| | | N (%) | P-value | N (%) | P-value | N (%) | P-value | N (%) | P-value |
| Agricultural, forestry, and fisheries | 80 | 2 (2.5) | | 0 (0.0) | | 0 (0.0) | | 0 (0.0) | |
| Production/technical | 294 | 10 (3.4) | | 3 (1.0) | | 1 (0.3) | | 0 (0.0) | |
| Soldier | 2 | 0 (0.0) | | 0 (0.0) | | 0 (0.0) | | 0 (0.0) | |

addition, significant differences in SFTS seropositivity were observed according to influenza vaccination. In Korea, the priority group for influenza vaccination is the elderly (> 65 years of age). In a previous study, influenza vaccination coverage in the general and high-risk groups was approximately 34.3% and 61.3%, respectively. Patients with chronic diseases, such as lung diseases, are mainly vaccinated against influenza virus [16].

In one case-control study involving 17,941 blood donors, 10 cases of multiple false-positives were observed, and 9 of these 10 were vaccinated against influenza virus. The mean time between blood donation and vaccination was 26 days, and approximately 0.6%–1.7% of blood donors vaccinated against influenza exhibited multiple false-positive ELISA results [17].

In our study, we observed a significant difference between the seroprevalence of SFTS with titers ≥ 1:128 and influenza vaccination, as well as a significant correlation between SFTS seroprevalence and underlying diseases (i.e., cardiovascular diseases). This SFTS seropositivity may be related to biological false positives due to influenza vaccination. Therefore, it is necessary to investigate the relationship between vaccination, underlying disease, and SFTS seroprevalence in future studies.

A few limitations must be considered when interpreting the findings of this study. Although samples were extracted from the total population, > 50% of the participants were residents of the Seoul and Gyeonggi regions; as such, most participants were from urban areas. Determining seroprevalence in rural areas is essential to assess the prevalence of SFTS. However, only 282 (18.8%) of the participants were residents of rural areas, which is significantly smaller than the 1218 (81.2%) participants from urban regions. Most importantly, the total number of participants nationwide in the 16 cities and provinces that underwent SFTS testing was only 1500, and such a sample size was not sufficient to represent the entire country. However, this study is meaningful because it was the first to investigate the SFTS antibody positivity rate in all 16 cities and provinces across Korea. Finally, due to the small number of seropositive samples, we could not conduct a multivariate analysis to identify factors related to seropositivity.

In conclusion, this study is the first to investigate SFTS seroprevalence in 1500 samples collected from 16 cities and provinces in Korea. The national SFTS antibody positivity rate was 3.7% using IFA IgG titer cutoff values ≥ 1:32 and 1.8% using an IFA IgG titer cutoff value ≥ 1:64.

## Supporting information

**S1 Fig. Representative indirect immunofluorescence antibody assay results showing the presence of SFTS-specific IgG antibody using serum samples collected in KNHANES.** A. IFA image of positive control; B. IFA image of negative control; C. Negative control image showing no staining; D. No.149 serum of the 1st batch (positive, image using serum with 1:32 dilution); E. No.108 serum of the 2nd batch (positive, image using serum with 1:32 dilution);

F. No.337 serum of the 2nd batch (negative, image using serum with 1:32 dilution).
(TIF)

**S2 Fig. Geographical distribution according to antibody titers of the 1500 participants by regions collected from the Korea National Health and Nutrition Examination Survey, 2014 to 2015.** The location of the dot showing the antibody titers in each region was randomly assigned in each region using by QGIS 3.26 program (https://qgis.org/ko/site/) and base map obtained from GEOSERVICE (Administrative Region of South Korea. http://www.gisdeveloper.co.kr/?p=8555#comments).
(TIF)

**S1 Table. The number of participants according to regions in this study, the population of each region, and the ratio of participants to the population (%).**
(DOCX)

**S2 Table. The number of participants according to age distribution and regions in this study, the population of each region, and the ratio of participants to the population (%).**
(DOCX)

## Author Contributions

**Conceptualization:** Choon-Mee Kim, Mi Ah Han.

**Data curation:** Mi Ah Han.

**Formal analysis:** Mi Ah Han, Na Ra Yun.

**Funding acquisition:** Dong-Min Kim.

**Investigation:** Mi-Seon Bang, You Mi Lee.

**Methodology:** Choon-Mee Kim, Mi-Seon Bang.

**Project administration:** Dong-Min Kim.

**Resources:** Na Ra Yun.

**Software:** Beomgi Lee.

**Supervision:** Na Ra Yun.

**Validation:** You Mi Lee.

**Visualization:** Beomgi Lee.

**Writing – original draft:** Choon-Mee Kim, Dong-Min Kim.

**Writing – review & editing:** Choon-Mee Kim, Dong-Min Kim.

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
