## [Decision Letter · Decision Letter 0]

12 Apr 2022

Dear Dr. Kim,

Thank you very much for submitting your manuscript "Seroprevalence of severe fever with thrombocytopenia syndrome using specimens from the Korea National Health & Nutrition Examination Survey" for consideration at PLOS Neglected Tropical Diseases. As with all papers reviewed by the journal, your manuscript was reviewed by members of the editorial board and by several independent reviewers. In light of the reviews (below this email), we would like to invite the resubmission of a significantly-revised version that takes into account the reviewers' comments. 

The Authors are expected to address all the criticisms by all Reviewers. In particular, please describe the representativeness of the samples, consider presenting the distribution of the titers (Reviewers #1 & #2), clarify and justify the cutoff titer for positivity (Reviewer #1), and describe clearly recruitment of the participants and sample collection, and the IFA method (Reviewer #2). In additional to the above comments, please address,

1. Table 3, results based on multiple cutoffs should be limited. It’s better to use a pre-defined or conventional cutoff for the main analysis.

2. Multivariable analysis can be done for if the number of positive cases are sufficient. Otherwise, descriptive analysis for some specific subgroups, e.g. by broad age groups and flu vaccination could be present.

We cannot make any decision about publication until we have seen the revised manuscript and your response to the reviewers' comments. Your revised manuscript is also likely to be sent to reviewers for further evaluation.

Sincerely,

Eric HY Lau, Ph.D.

Associate Editor

Nam-Hyuk Cho

Deputy Editor

The Authors are expected to address all the criticisms by all Reviewers. In particular, please describe the representativeness of the samples, consider presenting the distribution of the titers (Reviewers #1 & #2), clarify and justify the cutoff titer for positivity (Reviewer #1), and describe clearly recruitment of the participants and sample collection, and the IFA method (Reviewer #2). In additional to the above comments, please address,

1. Table 3, results based on multiple cutoffs should be limited. It’s better to use a pre-defined or conventional cutoff for the main analysis.

2. Multivariable analysis can be done for if the number of positive cases are sufficient. Otherwise, descriptive analysis for some specific subgroups, e.g. by broad age groups and flu vaccination could be present.

Reviewer's Responses to Questions

**Key Review Criteria Required for Acceptance?**

**Methods**

-Are the objectives of the study clearly articulated with a clear testable hypothesis stated?

-Is the study design appropriate to address the stated objectives?

-Is the population clearly described and appropriate for the hypothesis being tested?

-Is the sample size sufficient to ensure adequate power to address the hypothesis being tested?

-Were correct statistical analysis used to support conclusions?

-Are there concerns about ethical or regulatory requirements being met?

Reviewer #1: Short summary

The researchers have done an excellent job of estimating the seroprevalence of this disease. It is important to keep surveillance in order given the high CFR of SFTS. The current state of the manuscript needs to be improved before proceeding to publication. The points are described below. Main point is to present the data better, for example a better visualization of the representativeness of the sample and argument why a certain cut-off value was chosen. 

Major points

Line 80. Please provide estimates within the different risk groups or was it just one estimate for the whole group? I am curious to know what the seroprevalence is in the highest risk group or how much the seroprevalences variates from region to region. In line 84 it is suggested there are large differences between areas, please provide the numbers.

Line 110. The cut-off is determines who is positive and who is negative. The titers of 1:32 is based on results from 15 health check-up participants. The importance of the cut-off in this study warrants an elobation about how it was at 1:32 instead of referring to another study. 

Line 134 / Figure 1. A reason to do this study was to obtain representative estimates of the seroprevalence in South Korea. To underline this aim, it would be good to create a figure, where you show what fraction the number of participants are of the total population in each region. Likewise, would compare the age distribution in your sample with the age distribution in Korea. A good example is Figure in 1 in Backer et al. Eurosurveillance 2021 (Impact of physical distancing measures against COVID-19 on contacts and mixing patterns: repeated cross-sectional surveys, the Netherlands, 2016–17, April 2020 and June 2020). This provides information on the representativeness of your sample of the Korean population. 

Line 153. I think it would be great to show the distribution of the titers of the 1500 participants rather than mentioning the percentages in the text or shown in a table. The distribution can be shown with a histogram with the titers on the x-axis. 

Line 185 / Figure 3. I wonder whether the current representation is the best way. I am unsure what the message of the map is. Percentages are very hard to read. I would advise to use a choropleth map and choose one cut-off to be shown. This will probably underline the message that urban areas are less affected than rural areas.

Line 202 / Figure 4. I would focus less on finding a significant correlation and focus more on explaining / describing your data. Among those points is there any pattern in terms of rural areas vs urban areas? You could assess this by giving different colors to the points. Perhaps it reveals a certain pattern which tells you something about the surveillance of this disease. 

Line 202 / Table 2/3. Please provide a reason why you are looking at seropositivity at different cut-off values. To me your current approach is like P-value hacking, and it is probably better to stick to one cut-off. 

Line 277. You conclude that you observed a significant difference between SFTS and influenza vaccination, but for two cut-offs you did not observe this. 

Line 290. The manuscript would benefit from assessing how your estimate would change if your samples would be a better representation of the Korean population. With the Figure I suggested you to make, you’d have a good impression how representative your sample is in relation to age and region. Then you could elaborate how sure you are about your estimate of 3.7%, and in what direction you expect it go had you had a better sample. Or perhaps even better try to provide a weighted seroprevalence.

Minor points

Line 63. To an international reader the term 16 cities and provinces does not provide any additional context. Please specify whether the sample is representative of the country or not. 

Line 70: What does group 3 disease mean? Perhaps better to describe why it is categorized as a group 3 disease. Or describe to the reader why it matters. 

Line 194. I think you can remove “data not shown” here as well as in line 177. Data is mentioned in text.

Line 225 / Line 231. Please have a look at the titles of the tables. Title of table 2 is missing. 

Line 275. How related are SFTS and Influenza? It would be nice if this would be discussed in the discussion or introduction briefly.

Reviewer #2: (No Response)

**Results**

-Does the analysis presented match the analysis plan?

-Are the results clearly and completely presented?

-Are the figures (Tables, Images) of sufficient quality for clarity?

Reviewer #1: (No Response)

Reviewer #2: (No Response)

**Conclusions**

-Are the conclusions supported by the data presented?

-Are the limitations of analysis clearly described?

-Do the authors discuss how these data can be helpful to advance our understanding of the topic under study?

-Is public health relevance addressed?

Reviewer #1: (No Response)

Reviewer #2: (No Response)

**Editorial and Data Presentation Modifications?**

Reviewer #1: (No Response)

Reviewer #2: (No Response)

**Summary and General Comments**

Reviewer #1: (No Response)

Reviewer #2: Introduction

The introduction is too short and the similar work has already been done in some other countries, it could be great to develop the introduction with knowledge coming from other countries. There is even no information about the surrounding countries or even data in China, there have been several studies that have used the same study design as yours. Only from my point of view, the introduction should be rewritten.

Material and methods

1. The field study needs to be better explained as to how the sample had been collected and for what purpose? During which months were the samples collected?

2. An IFA image of a higher quality should be provided, also accompanied by a negative control image showing no staining.

3. There needs to be a more detailed description of the IFA method. At what concentration did you use the secondary antibody?

4. Statistics: A multivariate logistic might be better, as there are obviously collinearity between age and influenza vaccination. A trend analysis should be performed for the age comparison.

Results

1. How about the occupation of the subjects, and is there any difference for the IgG titier regarding their occupation?

2. The IgG positive sample should be shown for their neutralizing antibody.

3. Are there the simultaneously positive test from the same sample?

4. Lines 153: I can not understand the meaning of the sentences “The final antibody titer of the SFTS IFA IgM-positive sample was 1:64 (Table 1). Please rephrase.

5. Is there any qPCR test for those IgM positive samples?

6. You might need to list the months of the positive samples and compared with the negative ones.

Discussion:

You need to comment on how comorbidities could have contributed to the IgG positive detection.

The refences are not properly cited, for example, the discovery of SFTSV in China should be the NEJM paper, instead of the ref 2 (Kim KH, Yi J, Kim G, Choi SJ, Jun KI, Kim NH, et al. Severe fever with 307 thrombocytopenia syndrome, South Korea, 2012. Emerg Infect Dis. 2013;19(11):1892-4).

The identification of risk factors of seroprevalence is not refenced adequately, I would add reference in China (PMC4350533, Hu C et al；Ye X et al, PMC8600349) for the comparison with the risk factor related to antibody production in an endemic Region.

PLOS authors have the option to publish the peer review history of their article (what does this mean?). If published, this will include your full peer review and any attached files.

Reviewer #1: No

Reviewer #2: No
---

## [Decision Letter · Decision Letter 1]

7 Dec 2022

Dear Dr. Kim,

Thank you very much for submitting your manuscript "Seroprevalence of severe fever with thrombocytopenia syndrome using specimens from the Korea National Health & Nutrition Examination Survey" for consideration at PLOS Neglected Tropical Diseases. As with all papers reviewed by the journal, your manuscript was reviewed by members of the editorial board and by several independent reviewers. The reviewers appreciated the attention to an important topic. Based on the reviews, we are likely to accept this manuscript for publication, providing that you modify the manuscript according to the review recommendations. 

The Authors have addressed most of the comments but there are also some remaining issues raised by reviewer #2.

Sincerely,

Eric HY Lau, Ph.D.

Academic Editor

Nam-Hyuk Cho

Section Editor

The Authors have addressed most of the comments but there are also some remaining issues raised by reviewer #2.

Reviewer's Responses to Questions

**Key Review Criteria Required for Acceptance?**

**Methods**

-Are the objectives of the study clearly articulated with a clear testable hypothesis stated?

-Is the study design appropriate to address the stated objectives?

-Is the population clearly described and appropriate for the hypothesis being tested?

-Is the sample size sufficient to ensure adequate power to address the hypothesis being tested?

-Were correct statistical analysis used to support conclusions?

-Are there concerns about ethical or regulatory requirements being met?

Reviewer #1: (No Response)

Reviewer #2: (No Response)

**Results**

-Does the analysis presented match the analysis plan?

-Are the results clearly and completely presented?

-Are the figures (Tables, Images) of sufficient quality for clarity?

Reviewer #1: (No Response)

Reviewer #2: (No Response)

**Conclusions**

-Are the conclusions supported by the data presented?

-Are the limitations of analysis clearly described?

-Do the authors discuss how these data can be helpful to advance our understanding of the topic under study?

-Is public health relevance addressed?

Reviewer #1: (No Response)

Reviewer #2: (No Response)

**Editorial and Data Presentation Modifications?**

Reviewer #1: (No Response)

Reviewer #2: (No Response)

**Summary and General Comments**

Reviewer #1: I am happy with the authors response to the points raised in my initial review.

Reviewer #2: The authors had addressed most of the queries, however, some information should also be added to the text.

The authors should use appropriate statistic term, e.g.,Line 146, The chi-squared test or Fisher’s exact test was to assess the difference not the correlation between the investigated risk factors and SFTS.

Line 259, In addition, there was a significant difference between SFTS seroprevalence with titers ≥ 1:128 and influenza vaccination---please rephrase/

The chi-squared should be Chi-square

Distribution of Sample Characteristics should be Sample Characteristics or Distribution of Sample

PLOS authors have the option to publish the peer review history of their article (what does this mean?). If published, this will include your full peer review and any attached files.

Reviewer #1: No

Reviewer #2: Yes: WL

Figure Files:

Data Requirements:

Reproducibility:

References

---

## [Editor Report · Decision Letter 2]

13 Jan 2023

Dear Dr. Kim,

We are pleased to inform you that your manuscript 'Seroprevalence of severe fever with thrombocytopenia syndrome using specimens from the Korea National Health & Nutrition Examination Survey' has been provisionally accepted for publication in PLOS Neglected Tropical Diseases.

Best regards,

Eric HY Lau, Ph.D.

Academic Editor

Nam-Hyuk Cho

Section Editor

---

## [Editor Report · Acceptance letter]

9 Mar 2023

Dear Dr. Kim,

We are delighted to inform you that your manuscript, "Seroprevalence of severe fever with thrombocytopenia syndrome using specimens from the Korea National Health & Nutrition Examination Survey," has been formally accepted for publication in PLOS Neglected Tropical Diseases.

Best regards,

Shaden Kamhawi

co-Editor-in-Chief

Paul Brindley

co-Editor-in-Chief
